# A late B lymphocyte action in dysfunctional tissue repair following kidney injury and transplantation

Pietro E. Cippà[1,2], Jing Liu[1], Bo Sun [3], Sanjeev Kumar[1], Maarten Naesens[4,5] & Andrew P. McMahon[1]

The mechanisms initiating late immune responses to an allograft are poorly understood. Here we show, via transcriptome analysis of serial protocol biopsies from kidney transplants, that the initial responses to kidney injury correlate with a late B lymphocyte signature relating to renal dysfunction and fibrosis. With a potential link between dysfunctional repair and immunoreactivity, we investigate the immunological consequences of dysfunctional repair examining chronic disease in mouse kidneys 18 months after a bilateral ischemia/reperfusion injury event. In the absence of foreign antigens, a sustained immune response involving both innate and adaptive immune systems accompanies a transition to chronic kidney damage. At late stages, B lymphocytes exhibit an antigen-driven proliferation, selection and maturation into broadly-reacting antibody-secreting cells. These findings reveal a previously unappreciated role for dysfunctional tissue repair in local immunomodulation that may have particular relevance to transplant-associated immunobiology.

[1] Department of Stem Cell Biology and Regenerative Medicine, Eli and Edythe Broad Center for Regenerative Medicine and Stem Cell Research, University of Southern California, Los Angeles 90033-9080 CA, USA. [2] Division of Nephrology, Regional Hospital Lugano, Lugano 6900, Switzerland. [3] Molecular and Computational Biology, University of Southern California, Los Angeles 90089-2910 CA, USA. [4] Department of Microbiology and Immunology, KU Leuven, Leuven 3000, Belgium. [5] Department of Nephrology and Kidney Transplantation, University Hospitals Leuven, Leuven 3000, Belgium. Correspondence and requests for materials should be addressed to P.E.C. (email: pietro.cippa@eoc.ch) or to A.P.M. (email: amcmahon@med.usc.edu)

The immune system participates in tissue repair with contrasting effects: inflammation after injury is important to initiate the repair response but immune cells can contribute to secondary tissue damage[1–5]. Organ transplantation is an interesting model to investigate the interaction between tissue injury and local immune regulation. Ischemia/reperfusion injury (IRI) inevitably occurs during organ transplantation and triggers the coordinated activation of the innate and the adaptive immune system of the host in a complex immunological process leading to acute allograft rejection[6–8]. This process has been extensively investigated in experimental models and can be effectively prevented and treated with currently available immunosuppressive drugs[9].

The early and late immune responses to allografts are distinct processes: chronic forms of rejection remain mechanistically poorly understood and are not treated effectively[10–12]. As a result, the long-term outcomes after kidney transplantation have not substantially improved over 2 decades: approximately 4–5% of renal grafts are lost annually beyond the 1st year after transplantation, mainly because of late forms of immune-mediated injury (often referred to as chronic rejection)[12–14]. Refined pathologic and immunologic diagnostic tools (e.g., C4d stain and detection of anti-HLA antibodies) indicate a critical role for B lymphocytes and donor-specific antibodies in the late immune response to allografts[15,16]. However, at a time when the criteria for chronic antibody-mediated rejection are met, even aggressive immunosuppressive therapy does not substantially improve graft survival[10,17]. The mechanisms initiating a donor-specific immune response, several months/years after transplantation and often without any clear precipitating event, are unclear and clinically important.

In this study, we take advantage of recent technical advances in the characterization of the adaptive immune system and molecular processes determining the transition from acute to chronic kidney injury (CKI) to investigate the impact of dysfunctional kidney repair on the late immune response following kidney injury and kidney transplantation.

## Results

**Kidney injury and B lymphocytes after renal transplantation.** Transcriptional profiling of protocol biopsies from 42 kidney allografts in the 1st year after transplantation enabled an examination of changing gene activity in the post-transplant kidney. We identified a cluster of strongly correlated genes associated with fibrosis (e.g., COL1A1, DPT, MMP7), immunity (e.g., CD52, CXCL10, CCL21), and B cell action (immunoglobulin genes) (Fig. 1a). In agreement with the previous reports, B cell-associated transcripts and immunoglobulin genes increased over time in a subset of patients after transplantation (Fig. 1b–e)[18,19]. The heterogeneity of this population at 12 months after transplantation was highlighted by t-distributed stochastic neighbor embedding (t-SNE) analysis, which identified a group of patients separating from the remaining study population by virtue of the higher expression of genes associated to classical features of CKI, such as with fibrosis, wound healing and immunity (further identified as CKI-group; Fig. 1f–h, Supplementary Data 1)[1,20,21]. The sample cluster analysis based on B cell-associated genes indicated the subset of patients displaying a strong B cell signature substantially overlapped with the CKI-group (odds ratio for the overlap 45.5, $P < 0.0001$, Fisher's exact test, Fig. 1e). We reasoned that the strong concordance between chronic injury and B cell signature could be explained by two non-exclusive scenarios: (1) the host immune response to the allograft causes kidney damage and sustains the transition to CKI, (2) the B cell response is an intrinsic component of CKI, independently of alloreactivity.

**Kidney injury precedes B cell-mediated immunity.** To find evidence to distinguish between these alternative possibilities, we examined earlier biopsies from the patient cohort to examine the transcriptional response in the same kidney over time for factors that might precede the development of CKI and late B cell activity signatures. We compared the CKI-group (as defined at 12 months after transplantation, Fig. 1f) with the rest of the study population referred to as the "non-CKI" group. At baseline, clinical characteristics of the patients and transcriptional profiles of CKI and non-CKI kidney biopsies were indistinguishable (Supplementary Table 1). Three months post-transplant, renal function was similar between the two groups. No histological evidence was found for lymphocytic infiltrates compatible with rejection (similar t, i, and v scores according to Banff classification)[22] or chronic tissue damage (e.g., fibrosis as determined by the ci score) in the CKI group (Fig. 2a, b). Moreover, no patient developed de novo anti-HLA antibodies in the systematic screening at 3 months. However, CKI group patients expressed higher levels of well-characterized markers of acute kidney injury and repair (e.g., LCN2, SOX9, ALDH2A1; Fig. 2c)[23,24], suggesting a more pronounced or sustained response to tissue injury.

The percentage of patients with documented evidence for acute cellular rejection in the 1st year after transplantation in protocol biopsies or indication biopsies (including borderline changes) was similar among the groups (25% in the CKI and 29% in the non-CKI group). To find evidence for subclinical rejection episodes, we used an extensive list of genes associated with acute rejection that have been reported to detect rejection with greater sensitivity than conventional histology[25]. Among the 186 analyzed genes, only 10 were detectably up-regulated in the CKI group (Supplementary Data 2). These genes relate to innate immunity (Fig. 2d), whereas genes more specifically linked to rejection and adaptive immunity were similarly expressed in the two groups. In contrast, among a set of 29 genes previously associated with acute kidney injury in kidney transplant recipients[26], 19 (65%) were significantly up-regulated in the CKI group and only 1 in the rest of the study population (Supplementary Data 3). Thus, kidneys exhibiting a signature of fibrosis and B cell activity at 12 months showed evidence for kidney injury, but not allograft rejection at 3 months.

One year after transplantation, patients in the CKI-group displayed a lower renal function and higher scores related to chronic kidney damage compared to the non-CKI groups, consistent with transcriptional profiling (ci and ct score, Fig. 2e, f). In contrast, no histological evidence was observed for B cell-mediated immunity as a cause of chronic kidney damage (absence of peritubular capillaritis and C4d staining, Fig. 2g)[27]. Histological analysis at 12 months revealed no evidence of increased inflammation in the CKI-group, pointing to a higher sensitivity for transcriptome analysis. In summary, a transcriptional signature of acute kidney injury at 3 months was associated with chronic organ damage and lower eGFR at 12 months. The clinical data supported the hypothesis that fibrosis in CKI patients does not result from a pathological immune response to the antigenically distinct allograft. In contrast, the late immune response may reflect a dysfunctional immune response related to ongoing tissue injury.

**Ectopic lymphoid tissue following kidney injury in mice.** If late B cell activity is independent of alloreactivity, we would expect to find similar immunological processes during the transition from acute kidney injury to CKI. To explore this hypothesis beyond the descriptive association provided by the clinical data, we turned to a mouse model where IRI led to a dysfunctional repair process that transitioned to chronic kidney disease over months, to

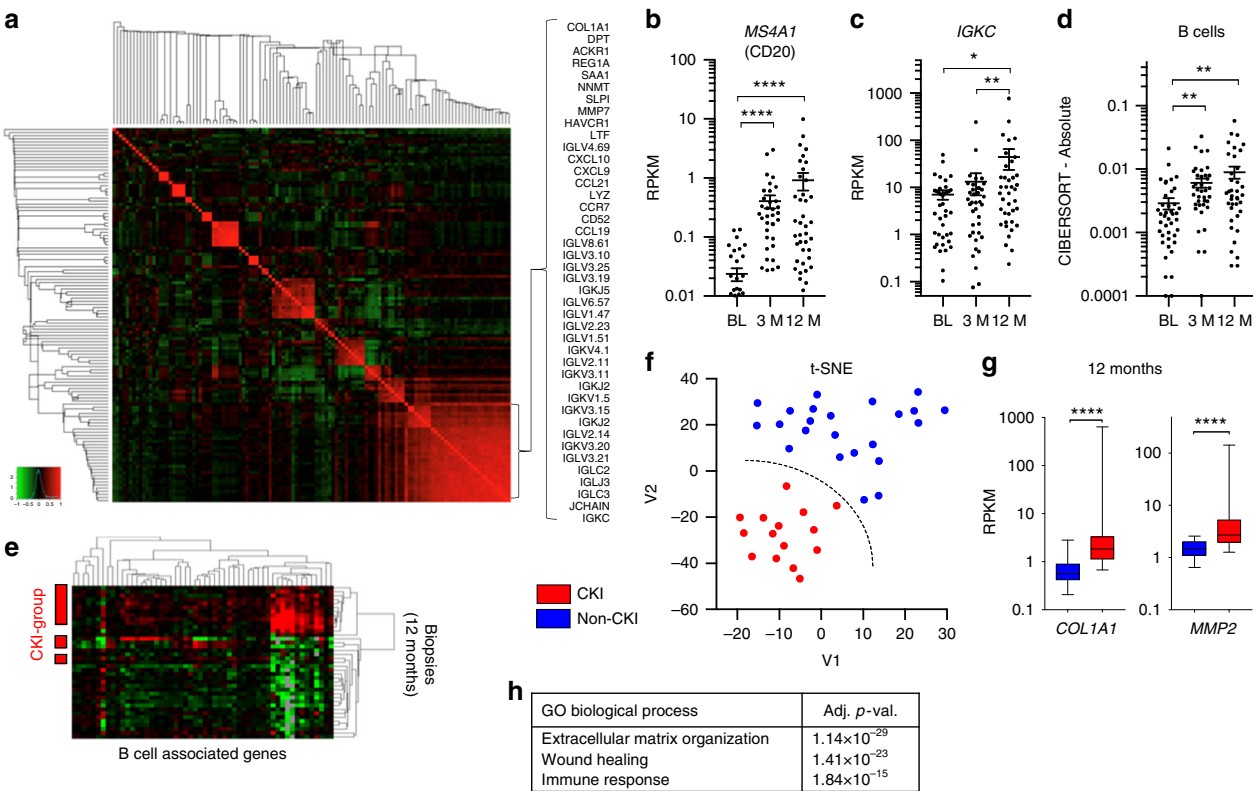

**Fig. 1** Chronic kidney injury and B cell immunity in human allografts. **a** Heatmap showing the gene expression correlation of the 120 most variably expressed genes across kidney biopsies collected at 3 and 12 months after transplantation ($N = 80$). The names of the genes included in the cluster in the bottom right corner are shown. **b**, **c** RPKM values of *MS4A1* (CD20) and *IGKC* over time in human kidney allograft biopsies. BL: baseline. $N = 39–42$ for each time point. Wilcoxon matched-pairs signed rank test, ****$P < 0.0001$, **$P < 0.01$, *$P < 0.05$. **d** Semi-quantitative evaluation of immune cell infiltrates in the kidney at different time points after transplantation as determined by CIBERSORT analysis on RNAseq data. $N = 39–42$ for each time point. Wilcoxon matched-pairs signed rank test, **$P < 0.01$. **e** Cluster analysis based on the expression of B cell-associated genes including kidney biopsies collected at 12 months after transplantation ($N = 39$). Patients classified in the CKI group are indicated on the left. **f** t-SNE analysis on RNAseq data from kidney biopsies collected 12 months after transplantation defining the classification of patients in the CKI group. The boundary was determined by visual examination of the t-SNE plot. $N = 39$. **g** RPKM values of *COL1A1* and *MMP2* shown as examples of genes differentially expressed in CKI and non-CKI. Mean value and standard error (SE) are shown. Mann–Whitney test, ****$P < 0.0001$. **h** Gene enrichment analysis including genes differentially expressed in the CKI group compared to non-CKI, $P$ values adjusted according to Benjamini–Hochberg

specifically characterize the long-term immunological sequelae of maladaptive kidney repair[28]. Sixteen months after a single IRI, the tissue architecture of the kidney was markedly distorted with flattened epithelial cells, cyst formation, extensive fibrosis and inflammation, reflected in a markedly reduced glomerular filtration rate (GFR) (Fig. 3a–e)[28]. CIBERSORT analysis on RNAseq data over time indicated that lymphocytes were the most abundant immune cells in the kidney beyond the 6th month after injury while myeloid cell expansion characterized the acute phase (Fig. 3f)[29,30]. Beyond the 6th month after injury, lymphocytes were mostly organized in large cellular clusters, primarily around small arteries, between renal tubules identified by *Havcr1* and *Krt20* activity, markers of unresolved tubular injury (Fig. 3g, Supplementary Fig. 1)[28]. These highly vascularized ectopic lymphoid structures were populated by CD19[+] B and CD3[+] T cells (mainly CD4[+]) and surrounded by Lyve1[+] lymphatic vessels.

Examining the cellular compartmentalization of larger cellular aggregates identified a B cell zone with germinal centers of CD19[+]/CD45R[+ or dim] B cells embedded in a network of CD21[+]/Cxcl13[+] follicular dendritic cells partially separated from clusters of highly proliferating Ki67[+] lymphocytes, as typically observed in mature germinal centers (Fig. 3h, i)[31,32]. Similar ectopic lymphoid structures were described in different clinical conditions in the kidney[33] and often develop in target tissues of autoimmune disease through the interaction of infiltrating immune cells and local stroma, directed by chemokine feedback loops[34,35]. A systematic analysis of the cytokines transcribed in the injured kidney over time highlighted the progression from acute to chronic inflammation, with the late detection of cytokines involved in the formation of ectopic lymphoid tissue (e.g., *Ccl21*, *Cxcl12*, *Cxcl13*; Fig. 3j)[34].

At 28 days post-IRI, the acute and the chronic inflammatory responses overlapped. To assess the contribution of different cellular compartments of the kidney at this critical transition point we performed RNA sequencing after Translating Ribosomal Affinity Purification (TRAP) to examine the profile of myeloid and renal stromal cell compartments (Fig. 3k)[36]. Both myeloid and stroma cells showed marked translation of cytokine-encoding mRNAs. However, the stroma cell compartment, which encompasses the bulk of cells directly contributing to renal fibrosis[37], expressed a cluster of cytokines showing elevated expression 6 months after IRI, including cytokines associated with lymphocyte homing (e.g., *Ccl28*, *Ccl21a*, and *Cxcl12*, Fig. 3j, k). Thus, T and B lymphocytes accumulate in the mouse kidney transitioning from acute to chronic injury in the absence of foreign antigens.

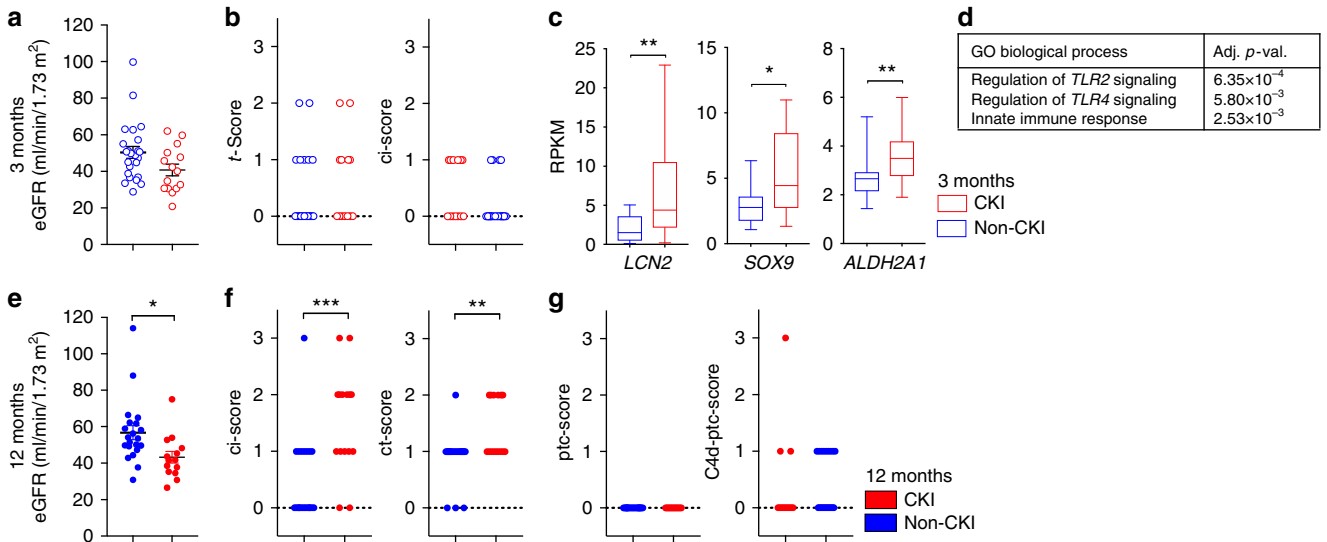

**Fig. 2** B cell action following tissue injury in human allografts. **a**, **e** Scatter plot showing the estimated glomerular filtration rate (eGFR) at 3 months (**a**, N = 41) and 12 months (**e**, N = 39) in kidney transplant recipients classified in the CKI and non-CKI group at 12 months. Unpaired t test, *P < 0.05. **b**, **f**, **g** Scatter plots showing the histological evaluation of protocol biopsies according to Banff at 3 months (**b**) and 12 months (**f**, **g**). t acute tubular lesions, ci chronic interstitial lesions, ct chronic tubular lesions, ptc peritubular capillaritis, C4d-ptc Cd4 positivity in peritubular capillaries. Mann–Whitney test, **P < 0.01. **c** Histograms showing mean RPKM values and SE of representative genes related to kidney injury and repair from kidney biopsies collected at 3 months after transplantation. Mann–Whitney test, *P < 0.05, **P < 0.01. Red symbols indicate patients of the CKI-group, blue symbols of the non-CKI group. Empty and filled symbols show data acquired at 3 and 12 months after transplantation, as indicated. **d** Gene enrichment analysis including genes with a higher expression in the MIR group at 3 months, among genes previously associated with acute rejection episodes. P values adjusted according to Benjamini–Hochberg

**B lymphocytes in the kidney transitioning to chronic injury.** The immune response detected in the kidney beyond the 6th month after IRI involved both T and B cells. The T cell compartment included a large fraction of non-conventional TCRαβ+ CD4−CD8− (double negative) T cells (Supplementary Fig. 2). Double negative T cells were predominant also among the lymphocytes isolated from the kidney of aged control mice, as shown in previous reports suggesting double negative T cells as a population of kidney resident T cells[38,39]. Consistent with the previous studies[5,40], B lymphocytes were rare in the normal kidney and memory B lymphocytes infiltrated the kidney in the first days after IRI; during the following weeks, B lymphocytes expanded massively and transcriptional analysis suggested a progressive switch to a plasma cell type (Fig. 4a, b). Infiltrating lymphocytes were isolated from kidneys 16–18 months after IRI by enzymatic tissue dissociation, magnetic cell sorting for CD45[41] cells and flow cytometry. FACS analysis confirmed the presence of differentiated B lymphocytes, but showed that the renal B lymphocytes lacked CD138 (Sdc1, syndecan-1, a typical plasma cell marker, Fig. 4c, d, Supplementary Fig. 3). Instead of fully differentiated plasma cells, we consistently identified a population of CD19+/lowCD45R− B lymphocytes, that displayed CD126 (interleukin 6 receptor) and Cxcr4 (CD184) (Fig. 4e, f, Supplementary Fig. 3), the receptor for Cxcl12, a cytokine highly expressed by the renal stroma in the transition to chronic injury (Fig. 3j, k).

B lymphocytes isolated from the kidney in the late phase after IRI were reminiscent of plasma cell precursors associated with renal production of autoantibodies in experimental lupus models and described in patients with systemic lupus erythematosus[42–44]. The marked increase in immunoglobulin gene transcripts for IgM, IgG2c, and IgA classes and Ig kappa light chains in whole organ RNA-profiles was consistent with the local production of antibodies in the kidney (Fig. 4g, h). Despite the high level of IgM transcripts and the recognized role of natural antibodies in the

tissue injury response[45], B-1 cells (and marginal zone B cells) were not detected in the kidney (Supplementary Fig. 4). Thus, antibody-secreting cells accumulated in the kidney in conjunction with the transition from acute to CKI.

**B cell clonal expansion and affinity maturation.** To better characterize the accumulation and the differentiation of B lymphocytes in the damaged mouse kidney we performed a B cell receptor (BCR) repertoire analysis in mice. First, we investigated the germline antibody gene segment repertoire by quantifying Ighv and Igkv transcripts in the kidney by whole organ RNA-seq. At 28 days after IRI, we observed the relative enrichment of a limited number of Ighv genes, reflecting the presence of a small number of B cell. In the following months, in parallel with the massive increase of immunoglobulin transcripts, we detected the relative enrichment of V segments. The 3 most prevalent V segments accounted for 25–40% of the total Ighv transcripts and the top one accounted for 18–35% of Igkv transcripts at 12 months (Fig. 5a–d). This process could only be appreciated beyond the 6th month after IRI, underlining the need for a very long-term follow-up, and consistent with the expansion of a subset of B cell clones in the damaged kidney.

The most frequent segments in both Ighv and Ihkv were present at high frequency in the 3 mice analyzed at 12 months, but were not particularly abundant in sham control mice and were not among the most frequently used V segments in mice[46]. This excludes an effect related to aging and suggests a biased usage of immunoglobulin genes. In a second step, the BCR repertoire analysis at 12 months after IRI was refined at the DNA level by Immunoseq, which better assays the composition of the B cell repertoire providing additional information on the single VDJ rearrangements[47]. This analysis confirmed the presence of a polyclonal B cell population in the kidney with the enrichment of a limited number of dominant clones. The amino acid sequences of the complementarity-determining region 3 (CDR3) of the

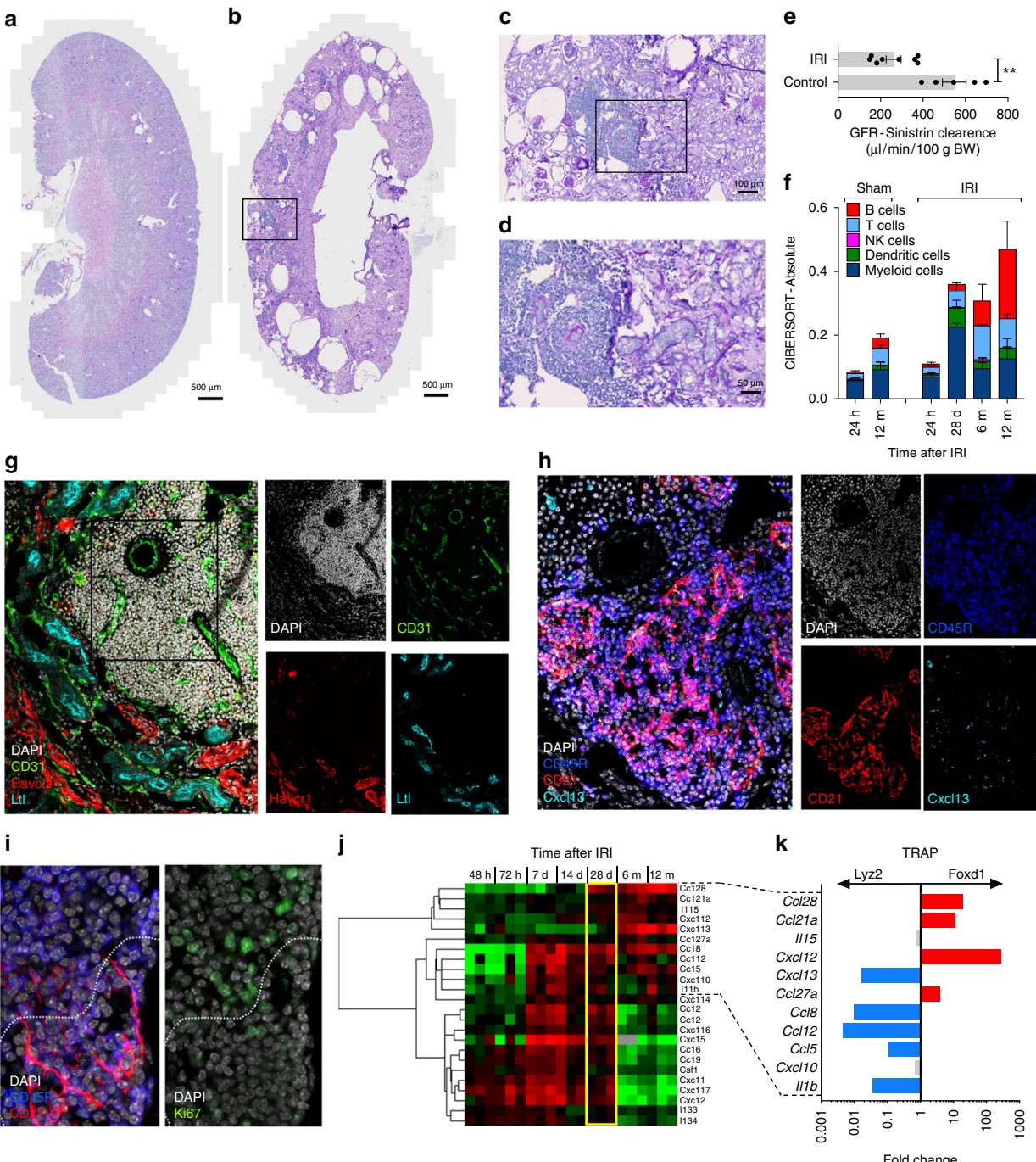

**Fig. 3** Tertiary lymphoid organs in the mouse kidney after ischemia/reperfusion injury. **a–d** Periodic acid-Schiff (PAS) staining of representative mouse kidney sections 16 months after IRI (**b–d**) or age-matched controls (**a**); $n = 3$/group; scale bar = 500 μm in (**a**, **b**), 100 μm in (**c**), and 50 μm in (**d**). **e** Glomerular filtration rate (GFR), as measured by sinistrin clearance, in 7 mice 16–18 months after IRI and in 5 age-matched controls. Mann–Whitney test, **$P < 0.01$. **f** Semi-quantitative evaluation of immune cell infiltrates in the kidney at different time points after IRI or sham surgery as determined by CIBERSORT analysis on whole kidney RNAseq data. Mean values and SE are shown, $n = 3$/group. **g**, **h** Immunostaining of consecutive sections obtained from a representative mouse kidney 6 months after IRI ($n = 3$–4/group). CD31: endothelial cells; Havcr1 (Kim1): marker of tubular injury; Ltl: proximal tubule. CD45R: B cells (and a subset of T cells, s. Supplementary Fig. 2); CD21 and Cxcl13: follicular dendritic cells. **i** Higher magnification of the B cell zone highlighting the partial separation of germinal centers in two areas: upper part with Ki67+ proliferating cells, lower part with CD21+ follicular dendritic cells. **j** Cluster analysis of cytokine transcripts obtained from RNAseq data from renal tissue at different time points after IRI ($n = 3$ for each time point). **k** Cell specific expression analysis of cytokine transcripts from RNAseq data obtained after TRAP from Foxd1-derived renal stroma cells and Lyz2-derived myeloid cells at 28 days after IRI. The ratio of the mean RPKM values (Foxd1/Lyz2) obtained from 3 independent mice, including cytokine genes involved in the late immune response according to panel (**j**) are shown. Genes specifically expressed in myeloid cells are shown in blue and in stroma cells in red

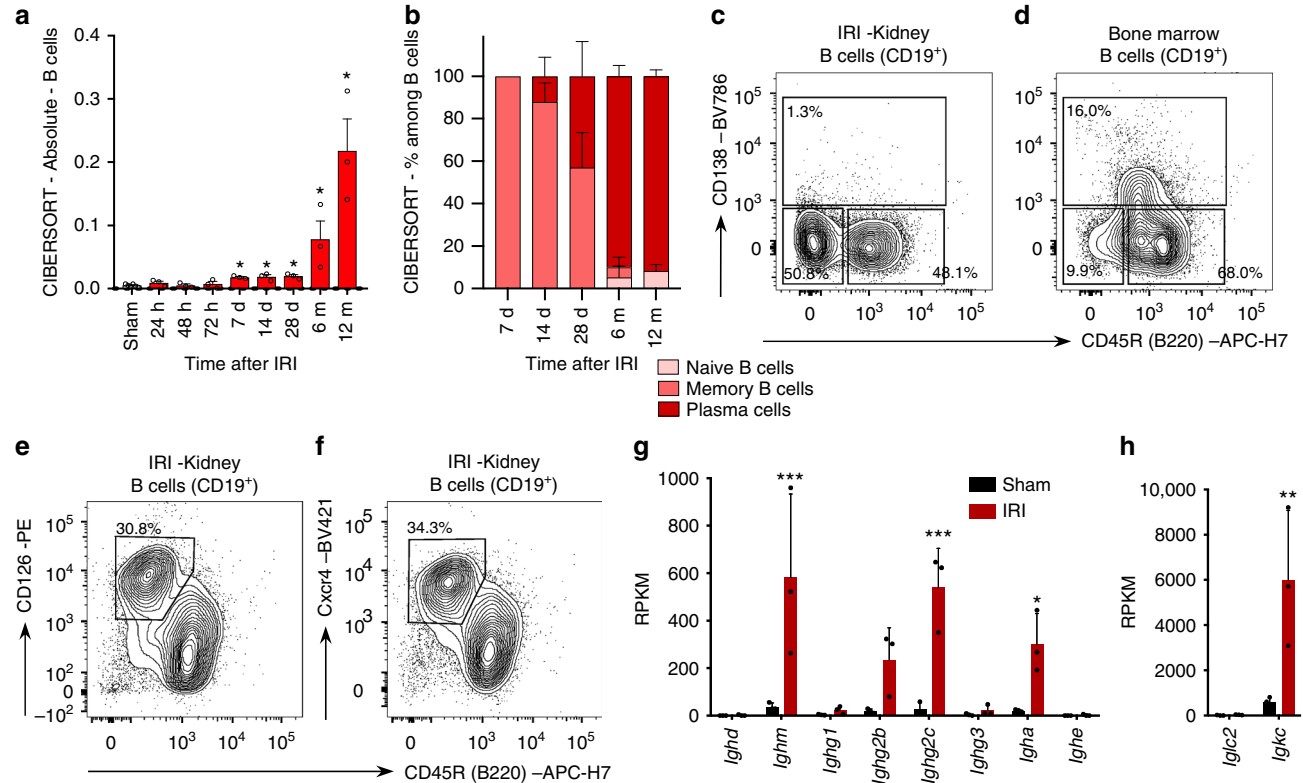

**Fig. 4** B lymphocytes in the late phase after ischemia-reperfusion injury in mice. **a, b** Semi-quantitative evaluation of immune cell infiltrates in the kidney at different time points after IRI or sham surgery as determined by CIBERSORT analysis on whole kidney RNAseq data ($n = 3$/group, Mann–Whitney test in comparison to sham control, *$P < 0.05$). **c–f** Flow cytometric analysis on CD45$^+$CD3$^-$CD19$^{+ \text{ or dim}}$ leukocytes isolated from the kidney or the bone marrow (BM) 16–18 months after IRI (1 representative example of 6 replicates is shown). General gating strategy is presented in Supplementary Fig. 5. **c, d** CD138 was absent on renal B cells, but detectable on the bone marrow B cells isolated from the same mouse (additional controls presented in Supplementary Fig. 3). **e, f** CD45$^+$CD3$^-$CD19$^{+ \text{ or dim}}$CD45R$^-$ B cells expressed high levels of CD126 and CD184 (*Cxcr4*). **g, h** RPKM values from RNAseq analysis on whole kidney 12 months after IRI or in sham controls focused on immunoglobulin constant regions transcripts ($n = 3$/group, Sidak's multiple comparison test, *adjusted-$P < 0.05$, ***adjusted-$P < 0.001$). In all histograms, mean values and SE are shown

immunoglobulin heavy chain in control mice did not show any evidence for a phylogenetic organization of the rearrangements (Fig. 5e). In contrast, IRI survivors presented clusters of hypermutated immunoglobulin gene rearrangements with few amino acid substitutions indicating a process of clonal expansion and affinity maturation (Fig. 5f). Thus, both histological findings and BCR analysis, provide evidence for the proliferation, selection and maturation of B lymphocytes in germinal centers within the kidney, in conjunction with the transition from acute kidney injury to CKI.

**Autoantibodies after ischemia-reperfusion injury.** The clonal expansion of B lymphocytes in germinal centers is an antigen-triggered immunological process[32]. In the absence of foreign antigens in this aseptic kidney injury model, we tested the plasma collected 16–18 months after IRI on a screening panel for autoantibodies. Thirty-five out of 118 tested antigens showed a statistically significant increased signal-to-noise ratio (SNR) 16 months after IRI compared to controls (adjusted $P$-value < 0.05, FDR 5%; two-stage step-up method of Benjamini, Krieger, and Yekutieli; Fig. 5g) without evidence for a common target antigen. This indicated that in the absence of foreign antigens, the intrarenal B cell response resulted in the production of broadly reactive autoantibodies. Since a similar process after kidney transplantation might stimulate the production of donor-specific antibodies, we retrospectively analyzed the clinical record of our study population to correlate the detection of anti-HLA

antibodies with the initial injury response: among kidney recipients not HLA-sensitized at time point of transplantation. Indeed, three patients developed clinical evidence for late B cell-mediated reactivity to the graft during further follow-up. All these patients were classified in the CKI group (Supplementary Table 2).

## Discussion

B lymphocytes are involved in the acute response to ischemic injury but are not essential for acute rejection, which is mainly mediated by T lymphocytes[3–5,48]. In contrast, B cells are pivotal in the pathogenesis of late forms of immune-mediated graft injury. Here, we show that late B cell activity in kidney allografts is tightly linked to dysfunctional kidney repair. Since in the clinical setting, this finding was associative, we turned to a mouse model to further characterize the immunological processes during the transition from acute kidney injury to CKI. Importantly, the aim of the experimental work in the animal model was not to mimic the clinical condition, but to examine the hypothesis formulated on patient data that B lymphocyte action is intrinsic to a late phase transition from acute kidney injury to CKI. A mouse model reliably replicating the transition to CKI following bilateral, warm ischemia-reperfusion served this purpose[28]. The findings were unexpected: in the late phase of the transition from acute kidney injury to CKI, similar B cell responses were observed to those from kidney transplant patient allograft biopsies,

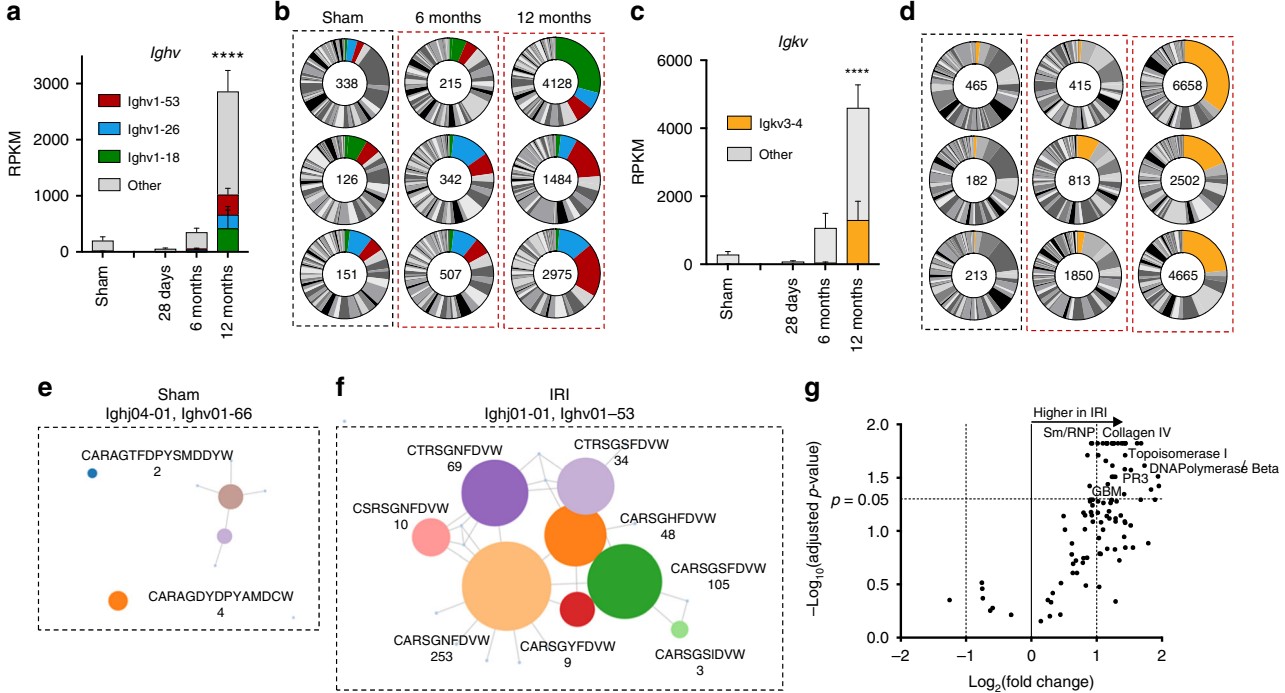

**Fig. 5** B cell receptor analysis and detection of autoantibodies. **a**, **c** Absolute RPKM values for germline *Ighv* and *Igkv* transcripts in 3 replicates collected 6 and 12 months after IRI and from sham control (matched to the 12 month IRI samples). The most frequent genes are highlighted with colors as indicated (mean values and SE are shown; $n = 3$/group; Mann–Whitney test, ****$P < 0.0001$ in comparison to sham control). **b**, **d** Relative frequency of germline *Ighv* and *Igkv* transcripts over time indicating the enrichment of common genes in the late phase; the number indicates the total RPKM of *Ighv* and *Igkv* at each time point. **e**, **f** Immunoseq analysis showing representatives cluster of vj rearrangements in sham and IRI mice (both 12 months after surgery). Each circle represents a unique amino acid sequence. The sequence is reported in the figure if >2 rearrangements were detected. The number of rearrangements is indicated below the amino acid sequence. The lines connect the circles if the corresponding rearrangements differ only in 1 amino acid. **g** Volcano plot comparison of signal-to-noise ratio in IgG reactivity in plasma samples collected 16–18 months after IRI ($n = 6$) and in age-matched controls ($n = 6$) on a microarray of autoantigens, with indication of log$_2$-fold change on the *x*-axis (>0 indicates higher in IRI) and significance level, expressed as log$_{10}$ of FDR-adjusted *P*-value. The name of representative antigens is indicated

suggesting that an antigen-driven immunological process is initiated in the absence of foreign antigens.

Though several elements in the transition from acute kidney injury to CKI are similar between mouse and man[49], the mouse model cannot be directly compared to the clinical setting of the transplant cohort due to fundamental differences in the type and kinetics of injury, and an absence of donor-distinct MHC antigens. Nevertheless, direct experimental evidence in the mouse IRI model suggests that in the absence of foreign antigens, a dysfunctional repair program transformed the mouse kidney into a tertiary lymphatic organ, not only hosting an inflammatory response typically associated with chronic organ injury[50], but also inducing an antigen-driven expansion and maturation of B cell clones. This process resulted in the production of broadly reactive autoantibodies, as previously reported in patients after heart and brain injury[51–53]. These findings are strikingly reminiscent of a recent characterization of the clonal evolution of autoreactive germinal centers in lupus. Once self-tolerance is broken in lupus, autoreactive B cells drive the expansion of further clones in the context of a self-sustained inflammatory feedback mechanism eventually leading to a convergent autoimmune response, several months after the initial break of tolerance[31]. Because of the high B cell precursor frequency against foreign MHC molecules, donor antigens are likely to contribute substantially, and to accelerate progression to this immunological process in the setting of transplantation.

Our findings support a model linking dysfunctional tissue repair in mouse IRI to inevitable tissue damage following organ transplantation in the clinic. In this model, B cell activity is linked to CKI and is not primarily a manifestation of alloreactivity. The maladaptive repair program following unresolved tissue injury is hypothesized to drive chronic activation of the adaptive immune system, including B lymphocytes. B lymphocyte interactions with the surrounding renal stroma likely create an environment supporting the recruitment of additional clones, stimulating maturation and differentiation of antibody-producing cells. Over time this process triggers the production of antibodies that promote further tissue damage in a deleterious feedback mechanism. In the presence of donor MHC antigens, this process is likely to activate donor-reactive B cells in the kidney and to trigger the production of donor-specific antibodies (Fig. 6). Further studies are required to specifically investigate how the presence of donor MHC antigens might influence the kinetics and the specificity of this immunological response. Furthermore, the pathogenicity of the autoantibodies detected in the mouse model needs to be determined because of the potential clinical relevance to kidney transplantation and to progression to chronic kidney disease after acute kidney injury.

Although the number of patients in the transplant cohort was not sufficient to demonstrate a causal link to donor-specific immuno-reactivity and to formally exclude potential confounders, the model is supported by earlier findings. Several studies characterized the broad antibody responses in conjunction with chronic rejection after kidney transplantation[54] and the acceleration of chronic rejection by intragraft lymphoid neogenesis[55]. Moreover, previous evidence for a stepwise breakdown of B

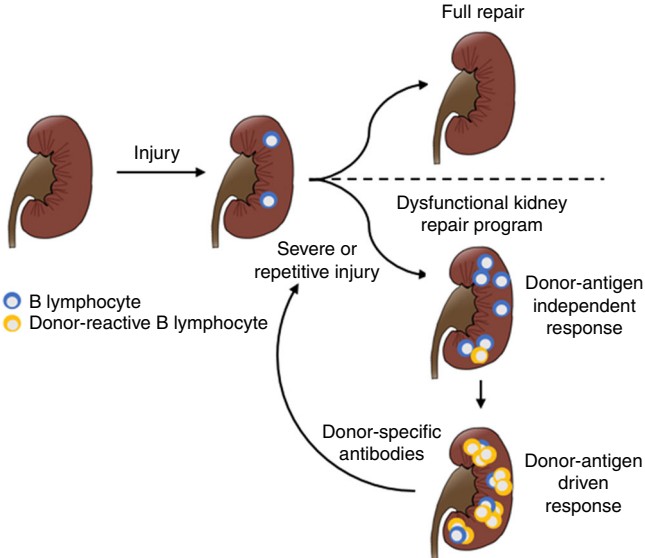

**Fig. 6** A new model to understand late immune responses to kidney allografts. Severe or repetitive kidney injury induces a dysfunctional repair program leading to a sustained immune response in the kidney. Over time, the local immune response leads to the recruitment and activation of donor-reactive B cell clones, which differentiate to plasma cells and produce donor-specific antibodies, further contributing to tissue injury in a deleterious feedback mechanism

cell tolerance as a preamble to chronic rejection directly parallels our observations in the mouse model[56]. Also consistent with this model are reports showing that among the anti-HLA antibodies detected 5 years following transplantation, only one-third were donor-specific, and non-donor-specific antibodies appeared earlier than donor-specific, but both donor-specific antibodies and non-donor-specific antibodies were associated with an adverse outcome[57,58]. Conversely, the identification of dysfunctional tissue repair as the initiator not only for fibrosis as previously proposed[19], but also for late forms of alloreactivity provides a plausible explanation for much lower incidence of chronic antibody-mediated rejection in an organ with a high regenerative capacity, such as the liver[11]. Although the current kidney study considers a quite limited number of patients in this cohort, with a low percentage of patients with clinical evidence for chronic rejection during follow-up, the findings presented support further validation considering independent study populations containing patients with elevated risk of organ rejection.

Our model argues for a change in perspective in relation to the unresolved issue of chronic allograft dysfunction and B cell-mediated immunity. In this, the production of donor-specific antibodies is a late event reflecting irreversible organ damage. This offers a potential explanation for a lack of efficacy in therapeutic approaches founded exclusively on suppressing chronic antibody-mediated rejection (including potent B cell depleting agents)[17]. Our data is in line with previous clinical studies[58] of a predictable path to chronic damage and B cell activation pointing to new opportunities for early diagnosis and novel therapeutic intervention. Clearly, injury in the clinical setting is much more variable than in the experimental mouse model. Several factors may determine whether sustained immune activation follows a renal repair response including age[59]. Developing strategies to attenuate the early injury response and repetitive hits during follow-up may prevent irreversible organ damage and late alloreactivity[60]. The process characterized here may also have broader significance in the pathogenesis of chronic organ failure in other clinical conditions[50,51].

## Methods

**Clinical study design**. We performed a genome-wide gene expression profile by RNA sequencing (RNAseq) in 42 kidney transplant recipients, randomly selected among patients with a full set of 3 available biopsies from a database at the University of Leuven. The patients received a kidney transplantation at the University of Leuven, Belgium. The study was performed according to all relevant ethical regulations, all patients gave written informed consent, and the study was approved by the Ethical Review Board of the University Hospital Leuven (S53364 and S59572). The renal biopsies were performed at the University Hospital of Leuven at following time points: before implantation (kidney flushed and stored on ice), 3 months and 12 months after transplantation (protocol biopsies). Additional biopsies performed in the same patients for a clinical indication were not considered for this study.

For histological evaluation, kidney sections were stained with hematoxylin eosin (HE), Periodic Acid-Schiff (PAS), and silver methenamine (Jones). All biopsies were centrally scored by pathologists dedicated to transplant pathology following the same standard procedures. The severity of chronic histological lesions was semi-quantitatively scored according to the Banff categories. The team involved in the computational analysis was not informed about any clinical information until the end of the study, when computational and clinical data were matched. The histological evaluation was independent of the computational analysis and the pathologist was not informed about the results of the transcriptional analysis.

**Human tissue storage and sequencing**. Of each renal allograft biopsy included in this study, at least half a core was immediately stored on Allprotect Tissue Reagent (Qiagen Benelux BV, Venlo, The Netherlands), and after incubation at 4 °C for at least 24 h and maximum 72 h, stored locally at −20 °C, until shipment to the Laboratory of Nephrology of the KU Leuven. We performed RNA extraction using the Allprep DNA/RNA/miRNA Universal Kit (Qiagen Benelux BV, Venlo, The Netherlands) on a QIAcube instrument (Qiagen Benelux BV, Venlo, The Netherlands). The quantity (absorbance at 260 nm) and purity (ratio of the absorbance at 230, 260, and 280 nm) of the RNA isolated from the biopsies were measured using the NanoDrop ND-1000™ spectrophotometer (Thermo Scientific™, Life Technologies Europe BV, Ghent, Belgium). Before library preparation, RNA integrity was verified by high sensitivity RNA ScreenTape analysis. Five samples were discarded because of not optimal RNA quality. Among the discarded samples, 3 were collected 12 months after transplantation, so that 39 patients could be classified in the CKI or non-CKI subgroups.

Library preparation and RNA sequencing were performed in two batches. The first batch consisted of a pilot study with 3 patients (9 samples) and the second batch included the rest of the study population. The library was prepared with Clontech SMARTer technology at the Genome Technology Access Center of the Washington University, St. Louis, MO. The sequencing was performed in the same lab by using the HiSeq 3000 system on the Illumina platform, with a target of 30 M reads per sample. The reads were aligned to the Ensembl top-level assembly with STAR version 2.0.4b. Gene counts were derived from the number of uniquely aligned unambiguous reads by Subread:featureCount version 1.4.5. Transcript counts were produced by Sailfish version 0.6.3. Sequencing performance was assessed for total number of aligned reads, total number of uniquely aligned reads, genes and transcripts detected, ribosomal fraction known junction saturation, and read distribution over known gene models with RSeQC version 2.3. All gene expression levels were normalized and quantified by RPKM (number of reads per kilobase per million mapped read).

**Mice and surgical procedures**. Mouse handling and husbandry and all surgical procedures were performed according to all ethical regulations for animal testing and research. The study received ethical approval by the Institutional Animal Care and Use Committees (IACUC) at the University of Southern California. Warm IRI was performed to induce ischemic acute kidney injury[36]: 10-to-12-week-old, 25–28 g, C57BL/6CN male mice, purchased from Charles River, were anesthetized with an intraperitoneal injection of a ketamine/xylazine (105 mg ketamine/kg; 10 mg xylazine/kg). Body temperature was maintained at 36.5–37 °C throughout the procedure. The kidneys were exposed by a midline abdominal incision and both renal pedicles were clamped for 21 min using non-traumatic microaneurysm clips (Roboz Surgical Instrument Co.). Restoration of blood flow was monitored by the return of normal color after removal of the clamps. All the mice received intraperitoneal (i.p) 1 ml of normal saline at the end of the procedure. Sham-operated mice underwent the same procedure except for clamping of the pedicles.

**Glomerular filtration rate in mice**. GFR was measured in mice by transcutaneous measurement of FITC-sinistrin disappearance with the NIC-Kidney device, purchased from Mannheim Pharma & Diagnostics GmbH (Mannheim, Germany)[61–63]. Mice were anesthetized with isoflurane. The back was shaved and carefully depilated with a depilation cream to optimize the contact of the optical components of the detector with the skin. The depilation cream was carefully removed and the detector was placed directly on the skin and fixed with a tape. With the mouse still anesthetized, FITC-sinistrin (ca. 7.5 mg/100 g body weight, purchased from Mannheim Pharma & Diagnostics GmbH) was injected into the tail vein. After the injection, the mouse was returned to its cage and transcutaneous detection of

FITC-sinistrin disappearance was measured for 45 min. The data were analyzed with the NIC-Kidney device related software, which automatically provides the FITC-sinistrin half-life. GFR was calculated as follows:

GFR [μl/min/100 g body weight] = 1416.8 [μl/100 g body weight]/$t_{1/2}$ (FITC-sinistrin) [min]

**Histology and immunofluorescence**. For conventional staining kidneys were perfused with ice-cold PBS and embedded in paraffin after overnight fixation in 4% paraformaldehyde (PFA) at 4 °C. Sections were cut at 2 μm and stained with hematoxylin and eosin, or PAS staining at the histology core of the USC Department of Pathology, Los Angeles, CA. For immunofluorescence, PFA fixed tissues were equilibrated in 30% sucrose/PBS overnight then embedded in OCT in dry ice ethanol bath. 8–10 μm frozen sections were washed in PBT (PBS + 1% Triton-X), blocked in 5% Normal Donkey Serum in PBT, and incubated overnight at 4 °C with primary antibodies and detected with species-specific secondary antibodies coupled to Alexa Fluor 488, 555, 594, and 647 (Life Technologies) for 1 h at room temperature. Following antibodies were used in this study to recognize CD3 (rabbit, Abcam, ab16669), CD19 (rat, Invitrogen, 13-0194-82), CD21 (rabbit, Abcam, ab75985), CD31 (rat, BD, 553370), CD45 (goat, BD, 550280), CD45R (rat, BD, 557390), Cxcl13 (goat, R&D, AF470), F4/80 (rat, eBioscience, 14–4801), Havcr1 (goat, R&D, AF1817), Ltl lectin-FITC conjugate (Vector Laboratories, FL-1321), Lyve1 (goat, R&D, AF2125), Ki67 (rabbit, Novocastra, Ki67p-CE). All sections were stained with Hoechst 33342 (Life Technologies) prior to mounting with Immu-Mount (Fisher). All images were acquired on Zeiss Axio Scan Z1 slide scanner and Zeiss LSM780.

**Lymphocyte isolation and flow cytometry**. To isolate kidney infiltrating lymphocytes the mouse was perfused with PBS until the kidney was visually blood-free. Then the kidney was removed, mechanically dissociated, and then incubated at 37 °C in digestion buffer (RPMI medium, 1 mg/ml type I collagenase, 10 μg/ml DNAse, 10% FCS, 25 mM HEPES, penicillin and streptomycin). After 60 min, the tissue was filtered through a 100 μm nylon mesh to remove remaining large fragments. CD45$^+$ cells were positively sorted by magnetic cell separation in an autoMACS with anti-mouse CD45 beads (Miltenyi Biotec, Cat# 130-053-301)[41]. CD45$^+$ cells were further characterized by flow cytometry (Aria II, BD Bioscience). Representative MACS sorting results and the general gating strategy are presented in Supplementary Fig. 5. Controls to exclude a bias related to CD45 MACS sorting are presented in Supplementary Fig. 3. Following antibodies were purchased from BD bioscience: BV421-1B1 (CD1d), FITC-17A2 (CD3), APC and PE-RM4-5 (CD4), BV786-53-7.3 (CD5), APC-53-6.7 (CD8a), PE-M1/70 (CD11b), BV605-HL3 (CD11c), FITC-1D3 (CD19), FITC and PE-1D3 (CD19), BV650–7G6 (CD21), APC-S7 (CD43), APC-IM7 (CD44), APC-Cy7–30-F11 (CD45), APC-H7-RA3-6B2 (CD45R/B220), PE-DX5 (CD49b), PE-MEL-14 (CD62L), BV421-H1.2F3 (CD69), PE-D7715A7 (CD126), APC and BV786-281.2 (CD138), BV421-2B11/CXCR4 (CD184), BV421-Jo2 (Fas), APC and BV605-11-26c.2a (IgD), BV605–11/41 (IgM), BV421-R2-40 (IgG2a/b), BV786 and BV650-PK136 (NK1.1), PE-J43 (PD1), BV421-H5–597 (TCRβ), BV786-GL3 (TCRγδ). Cell viability was assessed by fixable viability staining BV510.

**RNA sequencing of mouse renal tissue**. RNA was extracted from whole renal tissue or after TRAP[36], with a RNeasy kit (Qiagen) and provided to the USC Epigenome Center's Data Production Core Facility for library construction and sequencing. RNA integrity was verified by Bio-Rad Experion analysis. Library construction was carried out using the Illumina TruSeq RNA Sample Prep kit v2 through polyA selection. The manufacturer's protocol was followed with the exception that the final PCR amplification was performed for 12 and not 15 cycles. Libraries were visualized on the Agilent Bioanalyzer and quantified using the Kapa Biosystems Library Quantification Kit according to manufacturer's instructions. Libraries were applied to an Illumina flow cell at a concentration of 16 pM on a version 3 flow cell and run on the Illumina HiSeq 2000 as a paired-end read for 100 cycles each side. Image analysis and base calling were carried out using RTA 1.13.48.0. Final file formatting, de-multiplexing, and fastq generation were carried out using CASAVA v 1.8.2. The sequencing data were aligned to mm10 genome assembly with STAR aligner (version 2.5.0b). The mapping index was generated by GENCODE release M4 (GRCm38.p3) gene annotations. In addition to read mapping, STAR was also used to remove duplicates, generate read-count tables and wiggle files. Differentially-expressed genes were called for each time point with DESeq2. Cluster analysis of cytokine transcripts was performed with Gene Cluster (version 3.0) and visualized with Treeview (version 1.1.6r4).

**Autoantibodies**. Autoantibody reactivity against a panel of autoantigens was determined in mouse plasma samples at the UT Southwestern Medical Center. The samples were incubated with autoantigen array and the autoantibodies binding to the antigens on the array was be detected with Cy3 labeled anti-IgG and Cy5 labeled anti-IgM to generate Tiff images. Genepix Pro 6.0 software was used to analyze the image. SNR was used as a quantitative measure of the ability to resolve true signal from background noise. A higher SNR indicates higher signal over background noise. SNR equal or bigger than 3 was considered true signal from background noise. Quality control and normalization using variance stabilizing

normalization were performed according to the standard procedure at UT Southwestern Medical center.

**Statistical analyses**. RPKM and eGFR values were compared by $t$-test or by Mann–Whitney $U$ tests, a two-tailed $P$ value < 0.05 was considered significant. Categorical analyses were performed with Fisher's exact or Chi-square tests.

Differential gene expression analyses were performed with EdgeR[64], by applying an exact test, with a false discovery rate of 0.05. $P$ values were adjusted according to the Benjamini–Hochberg procedures. SNR values from the autoantibodies analysis were compared by the two-stage step-up method of Benjamini, Krieger, and Yekutieli with a false discovery rate of 0.05. Gene enrichment analyses were performed with ToppFun and $P$ values were adjusted according to Benjamini–Hochberg (https://toppgene.cchmc.org).

**CIBERSORT and Immunoseq**. CIBERSORT analyses were performed on RNAseq data by using the analytical tool developed by Newman et al. (cibersort.stanford.edu)[30]. LM22 was used as a reference signature for analysis on human data. The mouse-specific reference profile developed by Chen et al. was applied for studies in mice[29].

For Immunoseq analysis, mouse renal tissue was analyzed by Adaptive Biotechnologies (Seattle, USA)[47]. The data were analyzed with the immunoSEQ Analyzer (https://www.adaptivebiotech.com/immunoseq).

**Software**. Gene correlation analysis and feature correlation heatmaps were performed on the PIVOT platform, developed by the Kim Lab (http://kim.bio.upenn.edu/software/pivot.shtml)[65]. Dimension reduction analysis was performed with t-distributed stochastic neighbor embedding (t-SNE) by Rtsne package in R. Sample correlation analyses were performed on PIVOT[65] or on Cluster (version 3.0) and visualized in TreeView (http://jtreeview.sourceforge.net). Box plots, scatter plots, and histograms were generated with Prism 7. FACS data were analyzed with FlowJo 2, version 1.1.0.

**Reporting summary**. Further information on experimental design is available in the Nature Research Reporting Summary linked to this article.

## Data availability

The datasets generated and/or analyzed during the current study are available. Mouse RNAseq data are available on GEO as GSE52004, PMID: 24569379 and as supplementary table at https://doi.org/10.1172/jci.insight.94716. Human RNAseq data have been deposited as GSE126805.

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

## Acknowledgements

The authors thank Greg Alvarado, Jetty De Loor, Kari Koppitch, Gohar Seribekyan, and Eric Tycksen for technical support. Work in A.P.M.'s laboratory was supported by a grant from the California Institute for Regenerative Medicine (LA1-06536). P.E.C. was supported by the Swiss National Science Foundation (Grant 167773).

## Author contributions

P.E.C. and A.P.M. contributed to conception and experimental design. J.L. performed IRI surgeries. P.E.C., J.L. and S.K. performed experimental data acquisition and analysis. M.N. performed human data acquisition. P.E.C. and B.S. performed human data analysis. P.E.C. and A.P.M. prepared the manuscript, incorporating comments from other authors.
