## [Peer Review File · Nature Communications]

Reviewers' comments:

Reviewer #1 (Kidney disease, ischemia, inflammation)(Remarks to the Author):

Data from RNA sequencing of kidney allograft protocol biopsies (pre, 3 mo and 12 mo) obtained from 42 patients transplanted in Belgium (Dr Naesens), helped classify these patients into 2 subgroups. Group 1 (15 pts) had gene markers associated with fibrosis (e.g COL1A1, DPT, MMP7), inflammation (CD52, CXCL10, CCL21) and B cell activation (e.g Ig genes). Group 2 (24 pts) had no abnormal gene markers. Interestingly, Gp1 pts had similar clinical characteristics as Gp 2 pts i.e donor age, ratio of LD vs DBD vs DCD donors, allograft cold ischemia time (11 hrs), delayed graft function (10-15%) and HLA mismatches. Two pts in Gp 1 had antibody mediated rejection (see supplement data) while no pt in Gp 2 had rejection. Importantly, there was no difference in the genes linked to subclinical rejection at 3 months between the 2 groups. At 1 year, the mean GFR in Gp 1 pts was around 42ml/min while in Gp2 was 58 ml/min (Fig 2, $p < 0.05$). However, at 1 yr there was increased interstitial and tubular damage (Fig 2 f/g) in gp 1 pts even though there was no increase in inflammation between the two groups. Since there was no obvious clinical cause to explain the worsening GFR in Gp 1 pts and the genes associated with kidney injury (i.e 29 genes, see supp table 4) were increased at 3 months in Gp 1, the authors hypothesized that the kidney injury in Gp 1 resulted from the host immune response to the allograft as well as a maladaptive kidney injury repair response involving B cells that target the injured allograft. Hence Gp 1 patients were termed MIR (for maladaptive kidney injury repair) that worsens kidney function long term (12 months).

To prove their hypothesis they utilize a unilateral kidney ischemia (21 min) mouse model and follow these mice for 16 months. At 16 months (Fig 3), the IRI kidney is severely fibrotic, shrunken with cysts and is full of inflammatory cells (with lymphocytes > myeloid cells). The mean GFR of IRI kidney is around 275 ul/min/100g BW vs 525 ul/min/100g BW. The inflammatory response is predominantly lymphocytes at 6 months and B/plasma cells >T cells at 12 months. Plasma cells producing autoantibodies to a set of autoantigens (unclear if autoantigens are derived from kidney) predominate at 12 and 16 months. Many of these autoantibodies have mutated genes indicating that they may not be natural antibodies. Based on this unilateral IRI mouse model, the authors claim that B cells are involved in the maladaptive tissue repair in Gp 1 patients following kidney transplantation. Based on the current data and mouse model, the authors have not proved their hypothesis.

(1) Their mouse model provides an association between increased B/plasma cells and kidney injury. They would need to deplete the B cells post-injury (e.g with anti-CD20 antibody) to prove that B cells are involved in the subsequent fibrosis after injury. Secondly these autoantibodies may not be pathogenic even though they are mutated and lack certain characteristics of natural antibodies.

(2) In the mouse, predominant lymphocytic infiltration was seen at 6 months after ischemic injury, while in these patients there was no difference in histology at 12 months between Gp1 and Gp2 patients. Moreover, unlike the mouse model, biopsies at 12 months revealed no increase in inflammatory cells in Gp1 compared to Gp2 even though there was an increase in chronic renal damage on histology (fig 2 f/g, $p < 0.01$). The authors should provide an explanation.

Reviewer #2 (Kidney transplantation, tolerance induction)(Remarks to the Author):

Cippà and colleagues' manuscript offers an original mechanistic theory to account for chronic kidney allograft injury. The data are extensive and can be interpreted as supporting the authors' model; however, important questions remained unanswered. The editors will have to decide whether the original concept and experimental novelty out-weights the study's incompleteness. The main planks of the argument are:

- (1) Unresolved tissue damage following transplantation stimulates (in some cases) an on-going tissue-repair response involving innate immune and fibrogenic processes.
- (2) On-going tissue repair processes cause a local accumulation and non-specific activation of B cells within allografts through production of chemokines (eg. Ccl21).
- (3) B cells present in tertiary lymphoid structures in the kidney are somehow activated to produce a limited repertoire of antibodies, which might be primarily directed against auto-antigens.
- (4) The presence of ectopic germinal centers in allografts supports the development of graft-reactive antibodies, which sustain or exacerbate graft injury, which leads to irreversible damage.

Hence, the authors make a controversial claim that donor-specific antibodies are a consequence of so-called "maladaptive injury-repair" processes rather than being the initiating cause of chronic rejection. Further, although they do not explicitly say so, the authors' model does not invoke a specific T cell reaction, at least not to the point of producing donor-specific antibodies. Further, the authors seem to be claiming that (in some cases) donor-specific antibody responses are initiated within allografts, as opposed to lymph nodes or spleen. These are challenging ideas, but would have great implications if true.

The manuscript has two logically distinct parts: the human study is a hypothesis-generating step; the mouse part seeks to test these hypotheses. The human part is necessary to explain why the mouse experiments were done and, whatever criticisms are levelled at the human part, the authors cannot be disallowed their hypothesis that on-going injury precedes and elicits B cell responses. Hence, the manuscript's validity rests more on the mouse experiments. The authors chose to work in a warm kidney ischaemia-reperfusion model, not an allogeneic transplant model. This abstraction has a number of limitations, but most importantly, there is no alloantigen in the system, which could profoundly alter the dynamics of B cell responses. Based only on experiments in a syngeneic system, it is not fair for the authors to directly extrapolate their conclusions to allogeneic transplantation. Nevertheless, the data and underlying ideas presented in the manuscript are highly original and would undoubtedly stimulate much interest. If the editors look past its limitations and revise this article, please would the authors consider the following points:

1. Fig.1a legend – Does 'variably expressed genes' mean 'genes whose expression changed between m3 and m12'? Please clarify.
2. pg.2/l.59 – Did B cell-related transcripts increase in all patients equally, or more in the subset shown in Fig1e?
3. pg.2/l.64 – Please indicate whence the gene list associated with fibrosis, healing and immunity came.
4. Fig1f – How was the boundary determined? Is it arbitrary?
5. pg.2/l.66 – Calling this group "MIR" is drawing a functional conclusion from an unvalidated marker set. Do you find the same patient grouping based on a clinically validated gene set (eg. GoCar signature)?
6. pg.3/l.94 – At 3 months, do you have anti-HLA values to confirm lack of early sensitisation?
7. pg.3/l.96 – The mean eGFR in the MIR group doesn't appear to change between months 3 and 12, whereas there looks to be a slight improvement in mean eGFR in non-MIR patients. Does this improved eGFR in non-MIR recipients account for, "lower renal function" in the MIR group? Please show individual δ -eGFR in the two groups between 3 and 12 months.
8. pg.3/l.96 - Please show the number of acute cellular rejections and cases of de novo DSA in both groups at month 12. Judging from Table S3, there were only 3 de novo DSA in the MIR group? Has this number increased since writing this manuscript?
9. pg.3/l.102 – As point (7) the eGFR in the MIR group does not appear to change between 3 and 12 months, so it cannot be claimed that the transcriptional profile is associated with organ dysfunction.
10. pg.3/l.102-105 – It would be fair to say the clinical data led you to a hypothesis; however, the

stated conclusion is not justified by the results.

11. pg.3/l.114 – The cellular infiltrate at 6 months might have a completely make-up if this were an allogeneic transplant – especially if immunosuppression were required for the organ to survive that long. It is noted this is a model-specific description.

12. pg.4/l.129 – Are the up-regulated chemokines expressed in 28-day mouse kidney also found in the human dataset?

13. pg.4/l.138 – This conclusion seems to overdrawn because it seemingly implies a functional significance of the accumulation of B and T cells in injured organs – this is yet to be established.

14. pg.4/l.142 – What is the functional significance of the T cells? Do they sustain “maladaptive” tissue repair in this model? Are they necessary for B cell accumulation? Are they necessary for autoantibody production?

15. pg.5/l.175 – Again, the kinetics of this antibody response would presumably be very different in an allogeneic setting.

16. pg.5/l.200 – In the authors model, non-specific accumulation and activation of memory B cells in transplanted kidneys could predispose to de novo DSA formation. It is unclear to me whether the authors think that autoantibodies produced in the kidney might be pathologically relevant in the allogeneic setting. Have the authors transferred antibodies from 12 month IRI mice into healthy recipients to look for antibody-mediated renal injury?

17. In conclusion, what do the authors think is the relationship between B cell accumulation in IRI kidneys and on-going maladaptive repair processes? Do the B cells sustain the injury-repair process? If so, is this antibody-mediated, through production of fibrogenic mediators, or through stimulation of T cells? Has the experiment been performed with B cell- or antibody-deficient mice?

I greatly enjoyed reading this manuscript, which is highly original and describes some nice experiments. I doubt the relevance of the IRI model to allogeneic transplantation, but the underlying theory is very intriguing. The human data should not be too strongly interpreted (eg. p7.l252). I also encourage the authors to recognise the limitations of the mouse model and soften their conclusions (eg. p6.l228, p7.l248, etc.) accordingly. I am sure this article will stir-up much critical interest in the field.

Point-by-point responses to reviewers' comments

Reviewer #1

Data from RNA sequencing of kidney allograft protocol biopsies (pre, 3 mo and 12 mo) obtained from 42 patients transplanted in Belgium (Dr Naesens), helped classify these patients into 2 subgroups. Group 1 (15 pts) had gene markers associated with fibrosis (e.g COL1A1, DPT, MMP7), inflammation (CD52, CXCL10, CCL21) and B cell activation (e.g Ig genes). Group 2 (24 pts) had no abnormal gene markers. Interestingly, Gp1 pts had similar clinical characteristics as Gp 2 pts i.e donor age, ratio of LD vs DBD vs DCD donors, allograft cold ischemia time (11 hrs), delayed graft function (10-15%) and HLA mismatches. Two pts in Gp 1 had antibody mediated rejection (see supplement data) while no pt in Gp 2 had rejection. Importantly, there was no difference in the genes linked to subclinical rejection at 3 months between the 2 groups. At 1 year, the mean GFR in Gp 1 pts was around 42ml/min while in Gp2 was 58 ml/min (Fig 2, p <0.05). However, at 1 yr there was increased interstitial and tubular damage (Fig 2 f/g) in gp 1 pts even though there was no increase in inflammation between the two groups. Since there was no obvious clinical cause to explain the worsening GFR in Gp 1 pts and the genes associated with kidney injury (i.e 29 genes, see supp table 4) were increased at 3 months in Gp 1, the authors hypothesized that the kidney injury in Gp 1 resulted from the host immune response to the allograft as well as a maladaptive kidney injury repair response involving B cells that target the injured allograft. Hence Gp 1 patients were termed MIR (for maladaptive kidney injury repair) that worsens kidney function long term (12 months). To prove their hypothesis they utilize a unilateral kidney ischemia (21 min) mouse model and follow these mice for 16 months. At 16 months (Fig 3), the IRI kidney is severely fibrotic, shrunken with cysts and is full of inflammatory cells (with lymphocytes > myeloid cells). The mean GFR of IRI kidney is around 275 ul/min/100g BW vs 525 ul/min/100g BW. The inflammatory response is predominantly lymphocytes at 6 months and B/plasma cells >T cells at 12 months. Plasma cells producing autoantibodies to a set of autoantigens (unclear if autoantigens are derived from kidney) predominate at 12 and 16 months. Many of these autoantibodies have mutated genes indicating that they may not be natural antibodies. Based on this unilateral IRI mouse model, the authors claim that B cells are involved in the maladaptive tissue repair in Gp 1 patients following kidney transplantation.

We thank reviewer #1 for his summary of the logical steps behind this article, because it helped us to realize that some points required a more precise, unbiased explanation. The starting point of our study was the observation of the strong correlation between a signature of chronic kidney injury and B cell signature at 12 months after transplantation. This finding was not explained by the baseline characteristics of the grafts and patients. We reasoned that this correlation could be explained by two non-exclusive models: in the classical model, B cells would cause kidney injury leading to fibrosis in the context of late form of rejection; alternatively, we hypothesized that the B cell signature might not be related to allogeneic immunity but rather a manifestation of chronic kidney injury. The retrospective analysis of the transplant cohort was consistent with a predominant effect of the alternative model. Therefore, we established a mouse model to dissect the complex pathophysiology of chronic allograft nephropathy to specifically investigate donor-antigen independent elements. The results were unexpected: in the late phase of the transition from acute to chronic kidney injury we found a similar B cell response as in the allograft biopsies, consistent with an antigen-driven immunological process even in the absence of foreign antigens. The data presentation in the original version of the manuscript might not reflect our logical and unbiased approach. Therefore, we changed following aspects in the revised version of the paper:

- *We changed the name of the group originally termed MIR to the purely descriptive term "Chronic kidney injury", since MIR might suggest a biased approach.*
- *We substantially changed several paragraphs to better explain the logic of our work*
- *We explained the rationale for the mouse model, namely to investigate the transition from acute to chronic kidney injury in the absence of allo-immunity. Please note that – consistently with the idea of modelling AKI to CKD transition – we used a model of bilateral (not unilateral) kidney ischemia.*

Based on the current data and mouse model, the authors have not proved their hypothesis.

The main message of our paper is that B cell activity is an intrinsic component of the late phase of the transition from acute to chronic kidney injury, independently of any donor-specific antigen. Even in the absence of primary auto- or alloimmunity B cells accumulate in the damaged kidney and produce antibodies in an antigen-driven process. The data strongly suggest that B cell activity is not necessarily the “primum movens” of late forms of allograft rejection but provide the ideal immunological environment to trigger targeted immunological responses, which in the context of transplantation are very likely to include donor-specificity. Notably, the hypothesis motivating this study was not that late B cell activity after AKI is the cause of the transition from acute to chronic kidney injury (although we cannot exclude that this immunological response contributes to kidney injury). We designed this study to investigate the hypothesis that B cell activity is linked to chronic kidney injury and not primarily a manifestation of alloreactivity. The presented data provide good evidence that our main hypothesis is correct. The rationale of the study was explained better in the revised version of the manuscript.

(1) Their mouse model provides an association between increased B/plasma cells and kidney injury. They would need to deplete the B cells post-injury (e.g. with anti-CD20 antibody) to prove that B cells are involved in the subsequent fibrosis after injury. Secondly these autoantibodies may not be pathogenic even though they are mutated and lack certain characteristics of natural antibodies.

As mentioned above, the aim of this study was not to demonstrate that B cells are contributing to the transition to chronic kidney injury and there is no need to generate further evidence to demonstrate the critical role of B cells in late forms of allograft rejection (Loupy et al. N Engl J Med 2018). In fact, previous studies showed that B cells limit repair after IRI in the kidney (Jang et al., JASN 2010) and in the heart (Zouggari, Nat Med 2013). Despite strong evidence for B cells being involved in chronic forms of allograft rejection, the mechanisms triggering an immunological response several months after transplantation and often without any evidence for an “immunological danger” are poorly investigated. This project was designed to better understand the mechanism triggering the deleterious late B cell activity after kidney transplantation and highlighted that chronic kidney injury itself is the key. The discussion has been substantially revised to better explain these aspects.

Also if this was not the aim of the study, we agree with the reviewer that we cannot demonstrate that the autoantibodies detected in the mouse model are pathogenic. We tried to investigate if the autoantibodies detected in the serum deposit in the kidney by performing an anti-mouse-IgG and anti-mouse-IgM staining on kidney sections obtained 12 and 16 months after IRI. Unfortunately, the results cannot be interpreted since B6 mice spontaneously develop glomerular Ig depositions with age. We found a strong IgG and IgM positivity in the damaged kidney, but the staining was also positive in control aged mice.

(2) In the mouse, predominant lymphocytic infiltration was seen at 6 months after ischemic injury, while in these patients there was no difference in histology at 12 months between Gp1 and Gp2 patients. Moreover, unlike the mouse model, biopsies at 12 months revealed no increase in inflammatory cells in Gp1 compared to Gp2 even though there was an increase in chronic renal damage on histology (fig 2 f/g, $p < 0.01$). The authors should provide an explanation.

A direct comparison of the human and the mouse model is not appropriate. We designed the mouse model to study the transition from AKI to CKD and to mimic the transition to chronic kidney injury, as observed in a subset of kidney transplant recipients, but the two models are substantially different. Mice were exposed to one severe AKI event transitioning to CKD over time, whereas in kidney transplant recipients this process is likely to be the result of multiple hits (e.g. initial IRI, pharmacological toxicity, infections). The injury response characterized in the mouse model is much more severe and uniform than in the transition to chronic allograft injury occurring in patients. Therefore, it is not surprising that the massive lymphocytic infiltrates observed in

mice were not histologically detectable in humans. Nevertheless, at the transcriptional level we found a strong signature of chronic inflammation and B cell activity in association with chronic injury in humans (similar to the mouse model). These findings are consistent with the higher sensitivity of gene expression analysis compared to conventional histology in the analysis of kidney transplant biopsies (Modena et al, Am J Transplant 2016).

Reviewer #2

Cippà and colleagues' manuscript offers an original mechanistic theory to account for chronic kidney allograft injury. The data are extensive and can be interpreted as supporting the authors' model; however, important questions remained unanswered. The editors will have to decide whether the original concept and experimental novelty out-weights the study's incompleteness. The main planks of the argument are:

- (1) Unresolved tissue damage following transplantation stimulates (in some cases) an on-going tissue-repair response involving innate immune and fibrogenic processes.
- (2) On-going tissue repair processes cause a local accumulation and non-specific activation of B cells within allografts through production of chemokines (eg. Ccl21).
- (3) B cells present in tertiary lymphoid structures in the kidney are somehow activated to produce a limited repertoire of antibodies, which might be primarily directed against auto-antigens.
- (4) The presence of ectopic germinal centers in allografts supports the development of graft-reactive antibodies, which sustain or exacerbate graft injury, which leads to irreversible damage.

Hence, the authors make a controversial claim that donor-specific antibodies are a consequence of so-called "maladaptive injury-repair" processes rather than being the initiating cause of chronic rejection. Further, although they do not explicitly say so, the authors' model does not invoke a specific T cell reaction, at least not to the point of producing donor-specific antibodies. Further, the authors seem to be claiming that (in some cases) donor-specific antibody responses are initiated within allografts, as opposed to lymph nodes or spleen. These are challenging ideas, but would have great implications if true.

The manuscript has two logically distinct parts: the human study is a hypothesis-generating step; the mouse part seeks to test these hypotheses. The human part is necessary to explain why the mouse experiments were done and, whatever criticisms are levelled at the human part, the authors cannot be disallowed their hypothesis that on-going injury precedes and elicits B cell responses. Hence, the manuscript's validity rests more on the mouse experiments. The authors chose to work in a warm kidney ischaemia-reperfusion model, not an allogeneic transplant model. This abstraction has a number of limitations, but most importantly, there is no alloantigen in the system, which could profoundly alter the dynamics of B cell responses. Based only on experiments in a syngeneic system, it is not fair for the authors to directly extrapolate their conclusions to allogeneic transplantation. Nevertheless, the data and underlying ideas presented in the manuscript are highly original and would undoubtedly stimulate much interest.

We thank the reviewer for the positive evaluation of our work and for appreciating the novelty of our study.

We would like to precise some critical points:

- *The aim of the mouse model was to study the transition from acute to chronic kidney injury in the absence of foreign antigens. If the hypothesis generated in consideration of the clinical data was correct (i.e. unresolved tissue injury is intrinsically related to B cell activity), then we expected to find a B cell signature even in the absence of any allogeneic immune response. The experimental data demonstrated that our hypothesis was correct. We better clarified this aspect in the revised version of the manuscript.*
- *Ideally, to directly assess the effect of AKI to CKD transition with and without alloantigens we would have to perform additional experiments to compare IRI and transplanted kidneys in mice. In consideration of the long-term follow-up required to study the immunological processes of interest, these experiments would be extremely challenging and ultimately likely not informative: (1) kidney transplantation in mice is technically challenging, often leading to a non-optimal perfusion of the graft and the outcome is variable depending on the mouse strain combination. (2) Long-term immunosuppression therapy to prevent acute rejection episodes in mice is not established. (3) The multiple confounders influencing the outcome of a complex procedure like this would have required large groups of animals, an approach probably not acceptable in the current era of biomedical research. In consideration of the main purpose of using the mouse model, we believe that focusing on IRI to study the transition from acute to chronic kidney injury was sufficient to obtain the critical data to support our working hypothesis.*
- *The results we obtained from the mouse model were quite unexpected but at the same time eye-opening for us. The discovery of an antigen-drive B cell response within the kidney several months after IRI in*

correlation with the transition from acute to chronic kidney injury in the mouse and – in parallel – the strong correlation of B cell signature and chronic injury in human allografts changed our perspective on the pathogenesis of late forms of immunity in the transplanted kidney and supported the new model presented in this study. However, we agree with the reviewer that the last logical step in this model, namely the assumption that in the transplant setting a similar B cell response will lead to a donor-specific B cell response, is likely to occur but is not strongly supported by data presented in the manuscript. Notably, previous clinical studies support this new model. We better highlighted these aspects and explained the limitations of the study in the revised version of the manuscript.

- *Since B cells were the most prominent lymphocyte population in the long-term after IRI and antibody-mediated rejection is considered a main topic of interest in the current era of kidney transplantation, we decided to focus our experiments on the B cell compartment, but this does not mean that T cells are not important in this setting. More detailed considerations about T cells are reported below (comment #14).*

If the editors look past its limitations and revise this article, please would the authors consider the following points:

1. Fig.1a legend – Does ‘variably expressed genes’ mean ‘genes whose expression changed between m3 and m12’? Please clarify.

Genes included in figure 1a displayed the strongest variability among patients at 3 and 12 months. The figure legend was changed to better clarify.

2. pg.2/l.59 – Did B cell-related transcripts increase in all patients equally, or more in the subset shown in Fig1e?

B cell-related transcripts increased in a subset of patients only. This can be appreciated in figure 1b. Figure 1e highlights the fact that B cell associated transcripts were restricted to patients with chronic kidney injury (the MIR group in the original version of the manuscript). The text was slightly changed to avoid misunderstandings.

3. pg.2/l.64 – Please indicate whence the gene list associated with fibrosis, healing and immunity came. We performed a Gene Ontology analysis including the genes differentially expressed between MIR and non-MIR (reported as Suppl. Tab. 1) by using ToppGene, as indicated in the Methods. The genes displaying the strongest enrichment were associated with the terms indicated in figure 1h.

4. Fig1f – How was the boundary determined? Is it arbitrary?

The boundary was determined by the visual examination of the TSNE plot. A computational cluster analysis did not seem appropriate because of the relatively small number of samples.

5. pg.2/l.66 – Calling this group “MIR” is drawing a functional conclusion from an unvalidated marker set. Do you find the same patient grouping based on a clinically validated gene set (eg. GoCar signature)?

We agree that the group names might be misleading. As mentioned above (reply to reviewer 1), we decided to change the group names in the revised version of the manuscript to the more descriptive term “chronic kidney injury” CKI.

6. pg.3/l.94 – At 3 months, do you have anti-HLA values to confirm lack of early sensitisation?

Yes, anti-HLA were systematically measured at 3 months. No patient developed de novo anti-HLA antibodies at 3 months. All patients with anti-HLA antibodies at 3 months were sensitized before transplant. This was added to the text.

7. pg.3/l.96 – The mean eGFR in the MIR group doesn’t appear to change between months 3 and 12, whereas there looks to be a slight improvement in mean eGFR in non-MIR patients. Does this improved eGFR in non-MIR recipients account for, “lower renal function” in the MIR group? Please show individual δ -eGFR in the two groups between 3 and 12 months.

We thank the reviewer for this comment. The difference in eGFR among the two groups results indeed from an eGFR increase in non-MIR patients, which is not present in the MIR group. Interestingly, we found the same pattern in an independent clinical study (currently in revision in another journal): in this case, we determined eGFR slopes in kidney transplant recipients beyond the first year after transplantation. We found that eGFR improved in patients without any anti-HLA antibodies, whereas in patients with HLA-antibodies (both donor specific and non-donor specific) the slope eGFR was negative. As required by the reviewer, we show here the delta GFR at 3 and 12 months for each patient here, but we would not include this panel in the manuscript to avoid redundancy. [Redacted]

8. pg.3/l.96 - Please show the number of acute cellular rejections and cases of de novo DSA in both groups at month 12. Judging from Table S3, there were only 3 de novo DSA in the MIR group? Has this number increased since writing this manuscript?

The % of patients with acute cellular rejection (or borderline rejection) in the first year was similar in both groups: 25% in the “chronic injury” (MIR in the original version of the manuscript), and 29% in the remaining patients. In contrast, as presented in Table S3, all patients with de novo DSA were in the “chronic kidney injury” group. We verified the anti-HLA data on to the last follow-up: no other patient developed DSA in the following months. The low number of DSA is for sure not related to the lack of systematic screening, but rather reflect the low risk for long-term complications in this population of generally compliant patients, receiving a high-quality post-transplant care by a team of very dedicated nephrologists in the context of a healthcare system assuring universal cost coverage in Belgium.

The information related to acute cellular rejection episodes was added to the new version of the manuscript. Moreover, we mentioned in the discussion the need to validate the data in populations with higher risk for rejection in the new version of the discussion. In general, this additional data is consistent with the main message of the paper: cellular rejection should be considered as one of the possible causes of kidney injury after transplantation, but late B cell immunity seems to be independent of the initial type of injury.

9. pg.3/l.102 – As point (7) the eGFR in the MIR group does not appear to change between 3 and 12 months, so it cannot be claimed that the transcriptional profile is associated with organ dysfunction. “Organ dysfunction” was replaced by “lower eGFR”.

10. pg.3/l.102-105 – It would be fair to say the clinical data led you to a hypothesis; however, the stated conclusion is not justified by the results.

This paragraph was changed as follows:

“The clinical data supported the hypothesis that fibrosis in CKI patients might not result from a pathological immune response to the antigenically distinct allograft. In contrast, the late immune response might reflect a dysfunctional immune response related to ongoing tissue injury, not necessarily related to alloreactivity.”

11. pg.3/l.114 – The cellular infiltrate at 6 months might have a completely make-up if this were an allogeneic transplant – especially if immunosuppression were required for the organ to survive that long. It is noted this is a model-specific description.

We do not understand the reviewer’s point.

12. pg.4/l.129 – Are the up-regulated chemokines expressed in 28-day mouse kidney also found in the human dataset?

The results in cytokines were quite consistent between mouse and human. As shown in the plots below, most of the cytokines detected in the late phase in the mouse model were up-regulated in the chronic injury (previously MIR, here in orange) group compared to other patients (here in green) at 12 months. The data support the use of the mouse model, but – as explained in the reply to the last comment by reviewer #1 – a direct comparison of the mouse and the human model is not appropriate. Therefore, we prefer not to present these data.

13. pg.4/l.138 – This conclusion seems to overdrawn because it seemingly implies a functional significance of the accumulation of B and T cells in injured organs – this is yet to be established.

The sentence was changed to avoid any overinterpretation of the data.

14. pg.4/l.142 – What is the functional significance of the T cells? Do they sustain “maladaptive” tissue repair in this model? Are they necessary for B cell accumulation? Are they necessary for autoantibody production?

The first wave of lymphocytes accumulating in the kidney beyond the sixth month after IRI were T cells, and

T lymphocyte infiltration persist also in the later phase dominated by B cells. It is reasonable to consider T cells as functionally relevant in interaction with B cells in the formation of the intrarenal lymphatic structures and autoantibodies, but the experimental design does not provide any strong evidence to support this. As presented in Suppl. Fig. 2, intra-renal T cells in the late phase were mostly unconventional CD4⁺CD8⁻ (double negative) T cells, which might have peculiar functions in the local modulation of the immune response. The functional characterization of this intriguing lymphocyte population would require several new experiments with a very-long-term follow-up and is beyond the focus of interest of the present manuscript (s. above).

15. pg.5/l.175 – Again, the kinetics of this antibody response would presumably be very different in an allogeneic setting.

We agree that the kinetics would be different in an allogeneic model. This important consideration was included in the discussion, as follows: “The mouse model cannot directly compared to the clinical setting of the transplant cohort because of fundamental differences (involving the type of injury and the kinetics)...”

16. pg.5/l.200 – In the authors model, non-specific accumulation and activation of memory B cells in transplanted kidneys could predispose to de novo DSA formation. It is unclear to me whether the authors think that autoantibodies produced in the kidney might be pathologically relevant in the allogeneic setting. Have the authors transferred antibodies from 12 month IRI mice into healthy recipients to look for antibody-mediated renal injury?

This is an important point, which need to be clarified. The role of autoantibodies in chronic tissue damage has been previously investigated (e.g. Li et al., PNAS 2017), but to our knowledge their pathogenic effect is still very controversial. In the context of transplantation, the pathogenic effect of autoantibodies is probably much less relevant compared to the established deleterious effect of donor specific antibodies. The experiment suggested by the reviewer is potentially interesting, but unlikely to provide solid evidence for or against a pathogenic role of the autoantibodies in our model: the effect might be very modest and secondary effects related to serum injection might predominate on kidney pathology; moreover, the pathogenic effect of autoantibodies might depend on the exposure of antigens related to the response to injury, so that negative results would not exclude their pathogenic relevance. Therefore, we preferred to perform another experiment: to obtain indirect evidence of a potential pathogenic effect of the autoantibodies detected in the serum, which would require the detection of antibody deposition in the damaged kidney, we performed an anti-mouse-IgG and anti-mouse-IgM staining on kidney sections obtained 12 and 16 months after IRI (s. also reply to reviewer 1). Unfortunately, the results cannot be interpreted since B6 mice spontaneously develop glomerular Ig depositions with age. We found a strong IgG and IgM positivity in the damaged kidney, but the staining was also positive in control aged mice. Thus, the question about the pathogenic role of autoantibodies after transplantation remains open.

Nevertheless, our study indicates that the immunological response related to the transition from acute to chronic injury creates the prerequisites for a local antigen-driven B cell response. The link between this “unspecific” response (pathogenic or not?) and the donor-antigen specific B cell response after transplantation cannot not be demonstrated in our experimental model but seems very likely to occur in consideration of the well-known high precursor frequency in allogeneic immune responses. Moreover, this model is consistent with clinical studies indicating that non-donor specific anti-HLA antibodies can often be detected before DSA (and both are associated with a negative long-term outcome, Hourmant et al, JASN 2005).

17. In conclusion, what do the authors think is the relationship between B cell accumulation in IRI kidneys and on-going maladaptive repair processes? Do the B cells sustain the injury-repair process? If so, is this antibody-mediated, through production of fibrogenic mediators, or through stimulation of T cells? Has the experiment been performed with B cell- or antibody-deficient mice?

s. #16 and comments #1 by reviewer 1

I greatly enjoyed reading this manuscript, which is highly original and describes some nice experiments. I

doubt the relevance of the IRI model to allogeneic transplantation, but the underlying theory is very intriguing. The human data should not be too strongly interpreted (eg. p7.l252). I also encourage the authors to recognise the limitations of the mouse model and soften their conclusions (eg. p6.l228, p7.l248, etc.) accordingly. I am sure this article will stir-up much critical interest in the field.

We appreciate this positive comment by reviewer #2. The discussion was adapted to avoid any possible over-interpretation of the data, and to highlight the limitations of our study. We hope that this study will stimulate discussion and other innovative studies to improve long-term outcome after kidney injury and transplantation.

REVIEWERS' COMMENTS:

Reviewer #1 (Remarks to the Author):

The authors have clarified issues that I had with the manuscript

Reviewer #2 (Remarks to the Author):

This is an exciting story and I enjoyed reading your revised version. The changes answer most of my concerns. My only remaining difficulty is your justification of the mouse model. I understand the purpose of the abstraction, which was nicely described in your letter, and I fully appreciate the microsurgical problems of mouse kidney transplantation. However, critical readers could argue that everything you describe (broad-specificity abs, correlation to injury, intra-graft origin of responses, etc.) is **only** seen because you have no donor MHC antigen dominating the response. You leave readers to speculate what might happen if there were an alloantigen present: Would Ab responses initiate in lymphoid tissue instead of grafts? Would Ab formation occur faster than the injury response? Would allo-Ag change the cellular composition of the graft infiltrate? Maybe the whole description is model-specific? I feel that readers need to be led through this discussion. If you were more convincing about the relevance of your IRI model to allogeneic transplantation then your impact would be greater. To be clear, I am not suggesting that you should do transplant experiments for this article.

(4) A computational approach is not required; however, if you draw a subjective boundary on a scatter plot then you should say so in the legend. This is only one of many equally legitimate boundaries – why not a straight line that reclassifies the two patients with lowest V2?

(7) Great answer. Provided you are happy there is no selection bias then all good.

(12) My original question was really an attempt to find reasons to believe the autologous model is relevant to allogeneic setting. I am less concerned about comparing mouse and human than allogeneic to non-allogeneic. The data presented in your letter show comparability, right?

(16) Nice answer, but shouldn't you say a few words about the possible pathological relevance of auto-ab in your discussion?

This is a highly original piece of work. I have no further comments about the scientific content of the manuscript and recommend it be accepted.

Signed by James A. Hutchinson

Point-by-point responses to reviewers' comments

Reviewer #2 (Remarks to the Author):

This is an exciting story and I enjoyed reading your revised version. The changes answer most of my concerns. My only remaining difficulty is your justification of the mouse model. I understand the purpose of the abstraction, which was nicely described in your letter, and I fully appreciate the microsurgical problems of mouse kidney transplantation. However, critical readers could argue that everything you describe (broad-specificity abs, correlation to injury, intra-graft origin of responses, etc.) is **only** seen because you have no donor MHC antigen dominating the response. You leave readers to speculate what might happen if there were an alloantigen present: Would Ab responses initiate in lymphoid tissue instead of grafts? Would Ab formation occur faster than the injury response? Would allo-Ag change the cellular composition of the graft infiltrate? Maybe the whole description is model-specific? I feel that readers need to be led through this discussion. If you were more convincing about the relevance of your IRI model to allogeneic transplantation then your impact would be greater. To be clear, I am not suggesting that you should do transplant experiments for this article.

The discussion was substantially changed to better explain the strengths and the limitations of our experimental model.

(4) A computational approach is not required; however, if you draw a subjective boundary on a scatter plot then you should say so in the legend. This is only one of many equally legitimate boundaries – why not a straight line that reclassifies the two patients with lowest V2?

The figure legend was completed according to reviewer's recommendations.

(7) Great answer. Provided you are happy there is no selection bias then all good.

Thank you. We do not believe that any selection bias could influence the data here.

(12) My original question was really an attempt to find reasons to believe the autologous model is relevant to allogeneic setting. I am less concerned about comparing mouse and human than allogeneic to non-allogeneic. The data presented in your letter show comparability, right?

Yes, this is correct. We adapted the discussion to better address these aspects.

(16) Nice answer, but shouldn't you say a few words about the possible pathological relevance of auto-ab in your discussion?

We thank the reviewer for this suggestion. We added some comments on this to the new version of the discussion.

This is a highly original piece of work. I have no further comments about the scientific content of the manuscript and recommend it be accepted.

We thank the reviewer for the thorough and constructive evaluation of our work, which substantially improved the quality of the paper.